# Characteristics of Respiratory Syncytial Virus Infections in Children in the Post-COVID Seasons: A Northern Italy Hospital Experience

**DOI:** 10.3390/v16010126

**Published:** 2024-01-16

**Authors:** Davide Treggiari, Chiara Pomari, Giorgio Zavarise, Chiara Piubelli, Fabio Formenti, Francesca Perandin

**Affiliations:** 1Department of Tropical, Infectious Diseases and Microbiology, IRCCS Sacro Cuore-Don Calabria Hospital, 37024 Negrar di Valpolicella, Verona, Italy; fabio.formenti@sacrocuore.it (F.F.); francesca.perandin@sacrocuore.it (F.P.); 2Andrus Center, University of Southern California, Los Angeles, CA 90089, USA; chiara.pomari@icloud.com; 3Department of Pediatrics, IRCCS Sacro Cuore-Don Calabria Hospital, 37024 Negrar di Valpolicella, Verona, Italy; giorgio.zavarise@sacrocuore.it

**Keywords:** acute respiratory infection, respiratory syncytial virus, bronchiolitis, epidemiology, hypertransaminasemia, surveillance

## Abstract

Background: Public health measures for COVID-19 mitigation influenced the circulation of Respiratory Syncytial Virus (RSV) during the 2020–2021 winter season. In the following autumn, an unprecedented resurgence of RSV occurred. Our study monitored RSV pediatric infections one and two years after the relaxation of containment measures for the COVID-19 pandemic. Methods: We analyzed diagnostic molecular data for SARS-CoV-2, flu, and RSV infections and clinical data from children with respiratory symptoms referring to our hospital during the 2021–2022 and 2022–2023 seasons. Results: In the 2021–2022 season, the number of RSV-affected children was very high, especially for babies <1 year. The outbreak appeared in a shorter interval of time, with a high clinical severity. In the 2022–23 season, a reduced number of infected pediatric patients were detected, with a similar hospitalization rate (46% vs. 40%), and RSV accounted for 12% of the infections. Coinfections were observed in age <2 years. In RSV patients, symptoms were similar across the two seasons. Conclusions: The clinical presentation of RSV in the two post-COVID seasons suggests that the pathophysiology of the virus did not change across these two years. Further studies are needed to continuously monitor RSV to support an effective prevention strategy.

## 1. Introduction

Respiratory Syncytial Virus (RSV) is the most common cause of bronchiolitis, leading to hospitalization in infants worldwide and the second cause of infant mortality in resource-limited countries [1,2]. In particular, RSV clinically manifests with bronchiolitis in children younger than two years of age and especially in newborns under six months [3].

Despite the disease burden, no approved vaccines for RSV are currently available. The prevention of RSV infections in infants through maternal vaccination has become a priority and a target for the development of new potential vaccine candidates to be tested in clinical trials [4].

In addition, there are no specific therapeutic options for RSV which is due in part to our limited knowledge of the pathogenesis of the disease [1]. For this reason and the lack of specific etiological treatment, therapy is primarily supportive, based on oxygen and adequate fluid supplementation [5,6].

RSV is distributed worldwide and follows the pattern of influenza; despite strong seasonal trends, the circulation of RSV may increase in periods outside the typical season. Indeed, the surveillance system in Australia revealed a higher-than-expected number of cases during the austral spring of 2021, from September to December, while winter data indicated a lower-than-expected circulation of RSV [7]. These events were attributed to the impact of the COVID-19 pandemic and related containing measures, which deeply altered the natural course of seasonal viral infections [7,8,9].

Since the introduction of non-pharmaceutical interventions to control COVID-19, the circulation of RSV in Europe has been limited. Surveillance data from 17 countries showed delayed RSV outbreaks in France (≥12 weeks, w), the UK, and Ireland (≥4 w) during the 2020–21 season [10,11]. Cases of RSV (predominantly affecting young children) in France occurred in older children compared to previous seasons.

In Italy, restrictive measures for the prevention and control of the SARS-CoV-2 pandemic were likely to be responsible for the reduced circulation of RSV and other respiratory tract diseases during the 2020–2021 winter season, as detected by the Respiratory Virus Surveillance Network of the Italian National Institute of Health system (RespiVirNet, https://respivirnet.iss.it/Default.aspx?ReturnUrl=%2f, accessed on 15 November 2023). Although specific national data on RSV are not yet available, weekly surveillance reports from the Lombardia region confirmed a reduced circulation of the virus as well as a decrease in hospital admissions related to RSV bronchiolitis in that period. SARS-CoV-2 was the only virus detected in samples from individuals with influenza-like illness (ILI) tested between October 2020 and January 2021. These data confirmed the hypothesis that, as for other Countries, in Italy, the SARS-CoV-2 pandemic altered the RSV seasonality. As a result, in autumn 2021, we observed an unexpected surge in RSV infections among infants and elder children compared with previous seasons [12,13].

In this scenario, assays that can detect multiple respiratory viruses play a key role in both treatment decisions and infection control measures. In fact, they allow the monitoring of viruses in real-time as they move through the human population.

In this retrospective study, we evaluated the prevalence and clinical presentation of RSV infection among children referred to our hospital for respiratory problems during the 2021–2022 and 2022–2023 seasons. We compared the two periods, evaluating respiratory virus infection diagnostic and clinical data, including coinfections information, in order to evaluate the evolution of RSV pediatric infection in the post-COVID era.

## 2. Materials and Methods

### 2.1. Setting and Participants

This is a single-center retrospective observational study carried out in a hospital in the province of Verona, the IRCCS Hospital Sacro Cuore Don Giovanni Calabria in Negrar di Valpolicella (Verona, Italy). Inclusion criteria were all pediatric patients attending our hospital for suspected respiratory tract infections, with molecular analysis results of influenza virus (either A or B strain, INF-A/B), respiratory syncytial virus (either A or B strain, RSV-A/B), and SARS-CoV-2 test performed from 1 October 2021 to 31 January 2022 (2021–2022 season) and from 15 October 2022 to 31 January 2023 (2022–2023 season). Exclusion criteria were bronchiolitis or other respiratory inflammation with undetermined etiology and subjects transferred to other sites.

### 2.2. Ethics

The study was conducted according to the guidelines of the Declaration of Helsinki and approved by the Ethical Committee of Verona and Rovigo provinces under the protocol n. 30569 of 22 May 2023.

### 2.3. Molecular Analysis for SARS-CoV-2, Flu and RSV Infections

Nasopharyngeal swab samples were collected in viral transport media and analyzed with the Anatolia Geneworks Bosphore SARS-CoV-2/Flu/RSV PCR Panel Kit as per the manufacturer’s instructions. In total, 510 patients during the 2021–2022 season (from 1 October 2021 to 31 January 2022) and 365 patients during the 2022–2023 season (from 15 October 2022 to 31 January 2023) were subjected to the rt-PCR test.

### 2.4. Clinical Data Collection

All the patients attended the Pediatric Emergency Department and Operative Unit of Pediatrics of the IRCCS Hospital Sacro Cuore for lower respiratory tract infection symptoms. Clinical data were retrieved from the electronic medical records. The following clinical parameters were considered: length of hospital stay (days), the presence of dyspnea (arterial oxygen saturation SaO_2_ < 95%, in addition to respiratory discomfort such as nasal fin flaring, abdominal basculation, intercostal indentation), hypertransaminasemia (alanine aminotransferase, ALT > 40 UI/L), the duration of symptoms (referred to dyspnea, reported in days), aerosol therapy (with salbutamol or budesonide, using the pressurized metered-dose inhaler with valved holding chamber device), and the type of oxygen therapy (low flow if O_2_ < 2 L/min supplied by nasal prongs; high flow if O_2_ is supplied at ≥2 L/min by noninvasive oxygen therapy (NIV) using AIRVO Nasal High Flow system). Data were analyzed for each patient using Microsoft Excel 2013 software.

### 2.5. Statistical Analysis

All variables recorded in this study are presented using descriptive statistics. Continuous variables are presented as the mean ± standard deviation (sd) or median interquartile range (IQR 25th, 75th). Statistical significance was defined for a *p*-value equal to or lesser than 0.05 (Fisher’s test, Student’s *t*-test). Descriptive and association analyses were conducted using Microsoft Excel supported by the GraphPad statistics program.

## 3. Results

During the 2021–2022 season, 182 (36%) out of the 510 pediatric subjects (285 males and 225 females) were infected with a respiratory virus, 175 (96%) of whom had RSV-A/B and 7 (4%) had SARS-CoV-2. Seventy-five (41%) patients with lower respiratory tract infections were hospitalized in the pediatrics unit (46 males and 29 females, Table 1), all of them infected by RSV. The median age was 2.1 years.

During the 2022–2023 season, we found 105 (29%) subjects infected with a respiratory virus out of 365 analyzed subjects (206 males and 159 females). The median age for this period was 2.60 years. Differently from the previous season, influenza (72%) was the most accounted for infection, followed by RSV-A/B (11%) and SARS-CoV-2 (8%). Furthermore, during this season, we found different viral coinfections in 7 subjects, i.e., RSV-A/B and SARS-CoV-2 (4 subjects), RSV-A/B and INF-A/B (1 subject), and RSV-A/B, INF-A/B, and SARS-CoV-2 (2 subjects). Coinfections were observed in children <2 years of age. Fifty-two (49%) out of 105 infected subjects were hospitalized for a lower respiratory tract infection (32 males and 20 females, Table 1), among which 100% were RSV-infected patients. Even though it was a minor difference, the median age of the 2021–2022 season was significantly lower (2.15 years, IQR 1.00–5.00) compared to the 2022–2023 season (2.60 years, IQR 0.70–6.753, Fisher’s test *p* = 0.042). This difference increased considering only hospitalized patients, with a median age of 0.46 years (IQR 0.15–1.42) for the first season and 2.14 years (IQR 0.43–5.43) for the second season (*p* = 0.0001). Figure 1 shows the trend of RSV-A/B, INF-A/B, and SARS-CoV-2-positive children detected during the considered periods.

Focusing on the 2021–2022 season, the number of infected children increased gradually from October 2021 and reached a peak of 14 positive diagnoses per day around mid-November 2021. After this date, the numbers slowly declined, with less conspicuous peaks. From the end of the month, the decline was even more prominent, leading to few cases being detected in December. No additional new cases were detected in January 2022. Conversely, the RSV-A/B trend was much flatter during the 2022–2023 season, in which the maximum number of RSV-A/B positive cases was detected only at the beginning of December 2022 (4 cases per day), while the most prevalent infection was represented by INF-A/B, starting from October 2022 and spreading all over the season. All the positive flu cases were sent to the regional referral center for genetic characterization, and the most circulated influenza genotype was identified as A/H_3_N_2_.

We further analyzed the clinical characteristics of the hospitalized patients, and the data are summarized in Table 2. The following parameters were considered: the length of hospitalization, the presence of dyspnea, the presence of a high level of alanine aminotransferase (ALT > 40 UI/L), the duration of symptoms, and the type of oxygen therapy (low flow O_2_ < 2 L/min, or high flow with O_2_ ≥ 2 L/min administered by AIRVO).

We did not find any statistically significant difference in the average length of the hospital stay during the two seasons (mean ± sd, 5.0 ± 2.7 vs. 5.2 ± 2.7 days), even when considering only RSV-A/B inpatients. Moreover, we found that neither the type of virus nor the presence of coinfections had an impact on the duration of hospitalization during the 2022–2023 season (Table 2). Our data showed that dyspnea was present in 69% of cases in the first season; conversely, in the following season, only 40% of subjects showed this symptom (*p* = 0.0034). A significative difference was also found in the duration of symptomatology between the two periods, with a longer presence of symptoms observed in the 2021–2022 season compared to the 2022–2023 season (mean diff ± sd, 1.7 ± 0.7 days, *p* = 0.0255). When comparing the two seasons solely for RSV-A/B, we did not find significant differences in terms of both dyspnea (51 out of 75 during 2021/2022, 9 out of 12 during 2022/2023, *p* = 1.0, Fisher’s test) and symptoms duration (difference between means 1.2 ± 1.1, *p* = 0.307). No difference in terms of the presence of hypertransaminasemia was found between the two seasons (*p* = 0.713).

When analyzing the therapeutic interventions between the two seasons, we found statistically significant differences in aerosol (*p* < 0.0001), oxygen therapy with low flow (*p* = 0.0063), and AIRVO (*p* = 0.0044, Table 2), with a larger requirement of all the three therapies during the 2021–2022 season. For both seasons, almost all RSV children were given aerosol therapy (97 and 100% for the 2021–2022 and 2022–2023 periods, respectively), often because this therapy had already been started at home by the caregiver. Steroids are not indicated in pediatric guidelines, but in clinical practice, steroids are given to bronchiolitis patients for prompt improvement, favoring a positive recovery. No differences in steroid treatment were observed for RSV patients among the two seasons [14]. In the 2021–2022 season, 41 out of the 75 RSV inpatients required oxygen therapy (56%). Among patients receiving oxygen, twenty-two (53%) required the use of NIV at high flows with the aid of the AIRVO system for severe dyspnea. The mean duration of used high-flow oxygen therapy was 6 ± 2.4 days. The 22 subjects who needed high flow (29% of the RSV-A/B patients) were patients with a lower median age than subjects who needed low flow or no oxygen support. In addition, two of them were former preterm patients of 36 weeks (w) and 35 w ± 5 days. The latter subjects had a longer duration of high-flow respiratory therapy (12 and 13 days, respectively), with a longer duration of symptoms (19 days) and consequently longer hospitalization (12 and 13 days, respectively). In the 2022–2023 season, 31% of patients required low-flow oxygen therapy, and 10% needed AIRVO support. For both seasons, patients who needed high-flow oxygen therapy had a lower median age compared to all the hospitalized patients in the respective periods: 0.18 years (IQR 0.08–0.53) for the first season (−0.28 years compared to the median age of hospitalized patients in 2021–2022) and 1.94 years (IQR 1.72–7.80) for the second season (−0.20 years compared to the hospitalized patients in 2022–2023). This supports the hypothesis that younger babies are more subject to severe symptoms. Considering only the NIV-treated patients, the two seasons were statistically different (*p* = 0.004, Fisher test), but these data probably reflect the difference in the age of the hospitalized patients in the two seasons. However, no significant differences in terms of therapeutic interventions were found in RSV patients between the two seasons, with 58% of the 2022–2023 RSV patients requiring low flow O_2_ and 58% AIRVO. Ninety-nine percent (99%) of the reported RSV hospitalized cases manifested bronchiolitis, and only 1% manifested broncho-pneumonia, specifically a subject with comorbidity (pulmonary valve stenosis).

## 4. Discussion

RSV is one of the most common respiratory viruses. It affects not only young children but also the elderly and immunocompromised patients [15,16]. With the emergence of SARS-CoV-2, a considerable decrease in RSV incidence and hospitalization rate was observed worldwide during the 2020–2021 cold season [17,18,19,20,21,22], coinciding with the implementation of public health and social containment measures. A seasonality shift and a delayed RSV outbreak with a greater number of infected patients were reported in the 2021–2022 season in several countries, such as Australia, Saudi Arabia, New Zealand, France, the UK, and Japan [23,24,25,26,27,28]. Also, in Italy, an extraordinary surge in RSV was observed in the fall of 2021 [12,13,29], with hospitalization rates similar to the previous years but with a higher rate of admission to intensive care units [30]. Among others, the waning immunity against RSV in children and adults and a lack of antibody transmission from the mothers to the newborns seem to be the primary factors responsible for the exceptional resurgence of RSV in the 2021–2022 season [30]. This has led to increased attention towards RSV infections, which must be monitored to evaluate the evolving epidemiology and clinical manifestations. In our study, we assessed the clinical characteristics, in terms of symptoms and the subsequent healthcare management, of RSV pediatric infections in children admitted to our hospital one and two years after the relaxation of containment measures implemented during the COVID-19 pandemic.

In the first analyzed season (2021–2022), the number of RSV-affected children was very high, in line with literature data, especially among children under one year of age [31]. The outbreak appeared in a shorter interval, between October and December 2021, anticipating the expected epidemic by a couple of months and making it shorter. Pre-pandemic levels of infections were overcome, and, as a result, there was an overload in our hospital emergency rooms and the whole country [31]. A relevant feature of the 2021–2022 season was the severity of bronchiolitis: a more severe clinical presentation and a frequent need for high-flow oxygen therapy, especially in younger children compared to the previous years. In fact, in previous years, during RSV outbreaks in our hospital, it had never been necessary to use AIRVO for high-flow oxygen therapy and for such a prolonged time. There were significantly fewer cases of hospitalized children (lower than 30/years), with mostly home management. Unfortunately, because molecular testing was not often required in the pre-COVID-19 era and diagnosis was mostly clinical, we do not have sufficient data today to compare with what was observed in the post-COVID-19 era [29,31].

In the second analyzed season (2022–2023), a reduced number of infected pediatric patients were detected compared to the first post-COVID year (28% vs. 36%) but with a similar hospitalization rate (46% vs. 40%). During the second season, different viral infections were registered, with a predominance of the influenza virus (72%), while the RSV accounted for 12% of the detected infections. Another aspect that emerged in our study is the total absence of coinfections in the fall 2021 and their appearance in the fall 2022 season (7% of the total infected patients, 14% of the hospitalized). Coinfections were observed in children <2 years of age, suggesting that respiratory coinfections in children with SARS-CoV-2 are common, above all in younger children. This could be due to an immature immune response. A recent paper highlighted the role of interferons in preventing coinfections by different viruses [32]. In neonates and young children, the pathways involved in interferon production are still in a developmental state and can be less protective [33]. Chuang et al., reviewing the post-COVID pediatric infections data, reported a similar trend of SARS-CoV-2 and RSV coinfections in several regions [34]. The clinical effect of SARS-CoV-2 and RSV coinfection is still a debated issue; in fact, it seems to be correlated with a longer hospital stay, although a direct correlation with an increased mortality or intensive care unit admission has not been demonstrated [34,35].

Our study highlighted a decrease in the severity of symptoms among hospitalized pediatric patients in the second post-COVID year. Nevertheless, focusing only on RSV patients, a similar level of dyspnea, duration of symptoms, and need for oxygen therapy was observed, indicating that RSV clinical presentation was not changed across the two different seasons. Thus, the less severe symptoms registered globally in the 2022–2023 hospitalized population of pediatric patients were due to the presence of different viruses, such as SARS-CoV-2 and INFA/B.

The present study has the following limitations that should be considered. First, it was a retrospective analysis within a single hospital; testing was performed at the clinician’s discretion; clinical data were retrieved retrospectively, reading the letter of discharge from hospitalization; the data from the pre-COVID-19 era are missing. Moreover, we did not characterize the circulating RSV strains at the genomic level to evaluate possible differences between the two periods.

## 5. Conclusions

In this retrospective study, we pointed out that in the two years after the relaxation of the social containment measures, we observed a huge “pure” RSV outbreak in the 2021–2022 season, in which 96% of infected pediatric subjects carried RSV, followed by a more “mixed” 2022–2023 season, in which flu was prevalent, and RSV infections were reduced to 12%, including some coinfections. The clinical presentations of RSV in the two seasons were similar, suggesting that the pathophysiology of the virus has not changed across these two years. Further studies are needed to continuously monitor this virus to support the creation of an effective year-round RSV-specific prevention strategy and monitor the presence of respiratory viruses regardless of their seasonality.

## Figures and Tables

**Figure 1 viruses-16-00126-f001:**
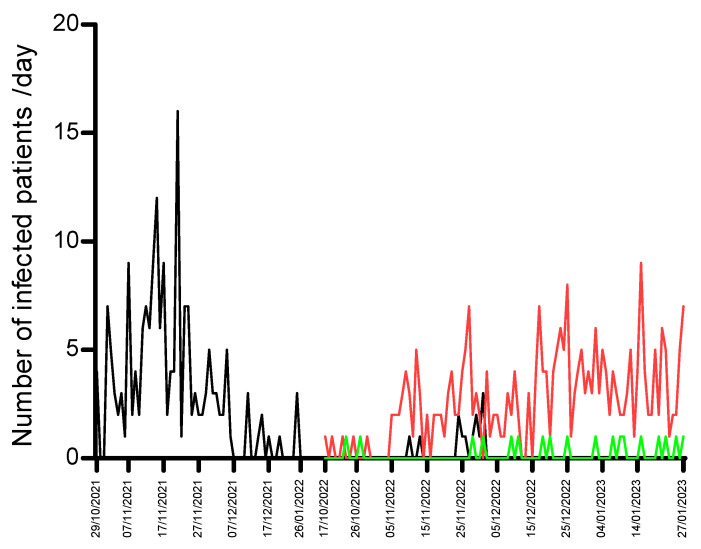
Epidemic trend of RSV, influenza, and SARS-CoV-2 infections from hospitalized children in our hospital during the seasons 2021–2022 and 2022–2023. The lines are represented by the following color code: Black line for RSV; red line for influenza; green line for SARS-CoV-2.

**Table 1 viruses-16-00126-t001:** Demographic characteristics of the study population. Population data were divided according to the analyzed seasons. The type and number of detected infections were reported. Lower respiratory tract infection was present in all hospitalized patients. *p* values for the Student’s *t*-test were reported.

		Season 2021/2022		Season 2022/2023		
Demographics		Count	Value (%)	Count	Value (%)	*p* Value
Population (n)		510		365		
Gender (n)						
	Female	225	44	159	44	
	Male	285	56	206	56	
Age (years)						
	25% Percentile	1.00		0.70		
	Median	2.15		2.60		0.0422
	75% Percentile	5.00		6.75		
**Infected (n)**		182	36	105	29	
	RSV	175	96	12	11	
	Influenza	0	0	76	72	
	SARS-CoV-2	7	4	9	8	
	RSV + SARS-CoV-2	0	0	4	4	
	RSV + Influenza	0	0	1	1	
	RSV + SARS-CoV-2 + Influenza	0	0	2	2	
**Hospitalised**		75	41	52	49	*p* value
Gender (n)						
	Female	29	38	20	38	
	Male	46	60	32	62	
Age (years)						
	25% Percentile	0.1		0.43		
	Median	0.4		2.14		0.0001
	75% Percentile	1.4		5.43		
Age (years) only for RSV infected						
	25% Percentile	0.15		0.06		
	Median	0.46		1.10		0.6396
	75% Percentile	1.42		2.38		

**Table 2 viruses-16-00126-t002:** Clinical characteristics of hospitalized patients. Data on hospital length and duration of symptoms were reported as mean ± standard deviation (sd) in days. Student’s *t*-test and Fisher’s test *p*-values were also reported, considering season 2022–2023 versus season 2021–2022 data—the duration of symptoms referred to the presence of dyspnea.

		Season 2021/2022		Season 2022/2023		
		Count	Value (%)	Count	Value (%)	*p*-Value
Number of patients (n)		75	41	52	46	
Infections (lower respiratory tract), (n)						
	RSV	75	100	12	23	
	Influenza	0	0	28	53	
	SARS-CoV-2	0	0	5	10	
	Co-infections	0	0	7	14	
Hospital lenght of stay (mean ± sd, days)						
	Overall	5.2 ± 2.7	-	5.0 ± 2.5	-	ns
	RSV	5.2 ± 2.7	-	5.0 ± 2.5	-	ns
	Influenza	0	-	4.7 ± 2.5	-	
	SARS-CoV-2	0	-	4.0 ± 2.5	-	
	RSV + SARS-CoV-2	0	-	6.5 ± 3.4	-	
	RSV + Influenza	0	-	4	-	
	RSV + SARS-CoV-2 + Influenza	0	-	6	-	
Clinical characteristics						
	Dyspnea (n)	52	69	21	40	0.0034
	ALT (>40 UI/L), (n)	7	9	4	7	ns
	Duration of symptoms (mean ± sd, days)	9.5 ± 0.4	-	7.8 ± 0.6	-	0.0255
Therapeutic interventions (n)						
	Areosol	73	97	32	62	<0.0001
	O_2_ (<2 L/min)	41	56	16	31	0.0063
	AIRVO (≥2 L/min)	22	29	5	10	0.0044
	Steroids	68	90	31	60	<0.0001

## Data Availability

The data supporting this study’s findings are available in the Zenodo Repository at https://doi.org/10.5281/zenodo.10159164 (accessed on 20 November 2023).

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
