# Peer review of "Characteristics of Respiratory Syncytial Virus Infections in Children in the Post-COVID Seasons: A Northern Italy Hospital Experience"

_viruses, 2024, doi:10.3390/v16010126_

Round 1

Reviewer 1 Report

Comments and Suggestions for Authors

The manuscript entitled “Characteristics of Respiratory Syncytial Virus Infections in Children in the Post-COVID seasons: a Northern Italy Hospital Experience” by Treggiari D et al. is a retrospective study evaluating the prevalence and clinical presentation of RSV infection among children admitted to an Italian hospital in Verona for the two post-COVID seasons (2021-22 and 2022-23). The authors concluded that the clinical symptoms of RSV in the two seasons analysed were similar suggesting that the pathophysiology of the virus did not change across these two years. Although the focus of the paper is RSV infection in children, the authors also included positive results and clinical data related to influenza and SARS-CoV-2 viruses in the study population. The topic of the paper is timely. However, the manuscript needs to be improved in terms of clarity and more interpretation of the results must be provided. Moreover, the conclusions of the study are not sufficiently supported by the statistical analysis given in the article and should be revised in the discussion. 

Comments:

Methods

  • Please check the start and end times of each season considered for the analysis. In the Paragraph 2.1 the authors reported the season 2021-22 from 1st October 2021 to 31st January 2022. However, in paragraph 2.3, they reported this season from 29th October 2021 to 25th December 2021.
  •  Considering the clinical manifestations, the authors added the duration of symptoms as a variable. What are the clinical symptoms considered (only dispnea and hypertransaminasemia)? It might be helpful to add the type of respiratory tract infection (upper or lower) in the table and in the description of the results. Did all hospitalised patients have a lower respiratory tract infection?

Results

· The presentation of the results needs to be improved. In the demographic characteristics of the study population, it might be helpful to add the age of the pediatric patients. Regarding this aspect, the authors reported that the median age of the 2021-22 season was significantly lower than the 2022-23 season. Is the median age reported in years or months? Please check the sentence and the reported data.

·  The description of the results regarding clinical characteristics should be revised. As the main objective and conclusion of the study are focused on RSV infection, the analysis and presentation of the results should be reported including only hospitalised patients with RSV infection. Information on the clinical manifestations of RSV-infected patients (bronchiolitis and broncho-pneumonia) should also be added to Table 2.

Comments on the Quality of English Language

 Minor editing of english language are required.

Author Response

We thank the Reviewer for taking the time to review this manuscript. Please find the detailed responses in the attached file and the corresponding revisions/corrections highlighted in track changes in the resubmitted manuscript.

Kind Regards

Chiara Piubelli

Reviewer 2 Report

Comments and Suggestions for Authors

The manuscript entitled: “Characteristics of Respiratory Syncytial Virus infections in 2 children in the post-COVID seasons: a Northern Italy Hospital 3 experience” is a single-center retrospective observational study, evaluating the prevalence and clinical presentation of RSV infection in the pediatric population in the seasons of 2021-22 and 2022-23. Like other reports, the authors showed a surge in number and severity of RSV cases in the season of 2021, that subsided in the season of 2022.

Overall, in this study the authors have compared the incidence and clinical severity of RSV infection and any other concomitant viral airway infection of children referred to tertiary Italian hospital one and two years after the ending of pervasive covid-19-restrictions.

The use of real-world data is a strength of the paper but the lack of information from the pre-covid-19 era is a limitation. The manuscript is easy for the reader to follow, but the manuscript needs some English editing. Please see my comments and suggestions:

1.     It is not clear whether the patients were admitted to hospital due to respiratory tract infection symptoms?

2.     Line 79, territorial(?) hospital

3.     What device was used for the aerosol therapy?

4.     What is the oxygene flow in the Optiflow device? Maybe add info in table 2.

5.     Consider skipping the text about most most appropriate descriptive statistics (line 109-110). Instead, add information to tables about what data that are presented (mean/median and sd/IQR) and what test that has been used. I think that Student’s test should be Student’s t-test or just t-test.

6.     This text on lines 132-133 doesn’t make any sense: Despite the small sample size, the median age of the 2021-22 season was significantly lower (5.57, IQR 1.86-17.07) respect to the 2022-23 season (1.08, IQR 0.29-3.10, p<0.001).

7.     Add age to table 1 and omit info about hospitalized patients in table 1.

8.     Add age to table 2.

9.     Use another wording than hypertransaminasemia. Maybe include the concepts high and low flow O2-therapy. Add SD/IQR of ALT-levels and duration of symtoms.

10.  What was the indication for steroids? What about comparing patients treated with steroids, were patients in the season 21/22 still more severely affected?

11.  Line 192, important? Dyspnea

12.  How much younger were the patients that needed NIV than those that didn’t 2021/2022? How did this compare to the need of NIV in 2022/2023?

13.  Is it possible to retrieve data on NIV treatment due to respiratory infection in the years before the covid-19 pandemic to estimate the difference of severely ill children? It is now only briefly mentioned in the discussion (line 229-231) without reference. Also statements line 234-236 should be supported by some data.

14.  Were there any differences in the circulating RSV viruses between the two studied periods? Could this have any impact on the results?

15.  Lines 246-255, are hard to follow, readability needs to be improved..

Comments on the Quality of English Language

There are some examples of odd wording in the manuscript. Consider editing by a native English speaker.

Author Response

We thank the Reviewer for for taking the time to review this manuscript. Please find the detailed responses in the atteched file and the corresponding revisions/corrections highlighted in track changes in the re-submitted file.

Round 2

Reviewer 2 Report

Comments and Suggestions for Authors

The authors have addressed my main concerns about the manuscript, but I think table 1 still is a bit confusing.

1.     In the legend past- and present tense are mixed.

2.     I can’t really get my head around how Fisher’s test can be used to compare ages between groups, thinking it is mainly a test for categorial data. Why not use t-test or Mann-Whitney test for differences in age? Or do the p values refer to any other differences between groups?

3.     Authors should also consider improving the lay-out of the table, since it is confusing with the header “Count (N)” for age.

Comments on the Quality of English Language

no comment
